# Expression of the Populus Orthologues of *AtYY1*, *YIN* and *YANG* Activates the Floral Identity Genes *AGAMOUS* and *SEPALLATA3* Accelerating Floral Transition in *Arabidopsis thaliana*

**DOI:** 10.3390/ijms24087639

**Published:** 2023-04-21

**Authors:** Xinying Liu, Qian Xing, Xuemei Liu, Ralf Müller-Xing

**Affiliations:** 1Institute of Genetics, College of Life Science, Northeast Forestry University, Harbin 150040, China; 2Lushan Botanical Garden, Chinese Academy of Sciences (CAS), Jiujiang 332900, China

**Keywords:** dual-function transcription factor, *YIN YANG 1* (*YY1*), gene duplication, REPO domain, flowering time, leaf curling, seed germination, root growth

## Abstract

*YIN YANG 1* (*YY1*) encodes a dual-function transcription factor, evolutionary conserved between the animal and plant kingdom. In *Arabidopsis thaliana*, *At*YY1 is a negative regulator of ABA responses and floral transition. Here, we report the cloning and functional characterization of the two *AtYY1* paralogs, *YIN* and *YANG* (also named *PtYY1a* and *PtYY1b*) from Populus (*Populus trichocarpa*). Although the duplication of *YY1* occurred early during the evolution of the Salicaceae, *YIN* and *YANG* are highly conserved in the willow tree family. In the majority of Populus tissues, *YIN* was more strongly expressed than *YANG*. Subcellular analysis showed that YIN-GFP and YANG-GFP are mainly localized in the nuclei of Arabidopsis. Stable and constitutive expression of *YIN* and *YANG* resulted in curled leaves and accelerated floral transition of Arabidopsis plants, which was accompanied by high expression of the floral identity genes *AGAMOUS* (*AG*) and *SEPELLATA3* (*SEP3*) known to promote leaf curling and early flowering. Furthermore, the expression of *YIN* and *YANG* had similar effects as *AtYY1* overexpression to seed germination and root growth in Arabidopsis. Our results suggest that YIN and YANG are functional orthologues of the dual-function transcription factor *At*YY1 with similar roles in plant development conserved between Arabidopsis and Populus.

## 1. Introduction

The genus Populus, which belongs to the Salicaceae or willow tree family, is of high commercial value [1,2,3]. Populus (*Populus trichocarpa*) features rapid growth, modest genome size, and established research methodology, and was the first tree genome completely sequenced, making Populus the most used model species for woody plants [4,5,6]. In plants, polyploidization is an important genetic phenomenon resulting in gene diversification and is the evolutionary origins of novel genes. The “salicoid” whole genome duplication (WGD) occurred 60–65 million years ago [6]. Within the 45,000 putative protein-coding genes identified in Populus, about 8000 pairs of duplicated genes have survived since the WGD event, which provide a greater genetic diversity [6,7].

In plants, flowering time is specified by the onset of flower production that is in some species accompanied by stem bolting. This change from vegetative to reproductive development plays an important role in the life cycle of plants. Flowering time has a deep impact on harvest time, yield, and quality of seeds and fruits [8]. Hence, manipulation of flowering time is a powerful tool to develop new plant varieties for crop production. The tendency of vegetative crops, such as carrots (*Daucus carota* subsp. sativus), onion (*Allium cepa*), and cabbage (*Brassica oleracea*), for early bolting and flowering upon exposure to low temperatures, thereby significantly reducing yield, have made resistance to bolting an important trait for breeding [9]. In contrast, artificial selection of more determinate growth habits and earlier flowering times provided more fruit in a shorter growing season in domesticated soybean (*Glycine max*), barley (*Hordeum vulgare*), wheat (*Triticum aestivum*), sunflower (*Helianthus annus*), and tomato (*Solanum lycopersicum*) [10]. In many trees and other woody perennials, breeding progress is hampered by the late onset of flowering [11].

As its full name *YIN YANG 1* implies, *YY1* encodes a dual-function transcription factor that can act either as activator or repressor [12,13]. First identified in the fly *Drosophila* and mammalians [12,14,15], YY1 is evolutionarily conserved between the animal and the plant kingdom [13]. YY1 belongs to the GLI-Krüppel family, containing four Cys2-His2 (C2H2) zinc fingers that form a DNA-binding domain [12]. *Drosophila* pleiohomeotic (PHO), which is highly homologous to mammalian YY1, functions as a Polycomb protein that epigenetically represses its target genes [15]. Mammalian YY1 participates in embryonic development and tumor growth regulation and may serve as a pioneer factor leading to the recruitment of other transcription factors and chromatin modifiers to activate its target genes [16,17]. The first study on YY1 in plants was performed on the C4 plant maize (*Zea mays*), in which YY1, here named transcription repressor maize 1 (TRM1), represses in leaf mesophyll cells *ribulose bisphosphate carboxylase S* (*rbcS*) that encodes the small subunit of a photosynthetic carbon dioxide fixation enzyme [18]. In Arabidopsis, *At*YY1 interacts with the mediator subunit MED18 to repress the disease susceptibility genes glutaredoxins *GRX480* and *GRXS13* and thioredoxin *TRX-h5* [19]. *At*YY1 acts a negative regulator of ABA responses by direct activation of *ABA REPRESSOR 1* (*ABR1*) via binding of a CCATnTT motif in the *ABR1* promoter [13]. Furthermore, *At*YY1 contains a casein kinase II (CKII) phosphorylation site (S284), while S284 phosphorylation enhances the transcriptional activity and protein stability of *At*YY1 and hence strengthen the effect of *At*YY1 as a negative regulator in the ABA response [20]. Recently, it has been shown that *At*YY1 associates, likely via its REPO-domain, with the chromatin-remodeling factor INO80 and the ATX4/5-containing COMPASS III complex that is involved in transcriptional activation via histone H3K4 trimethylation [21,22,23].

Flowering time is regulated by several pathways involving dozens of genes controlling the molecular mechanisms of floral induction in Arabidopsis [24]. The convergence of the individual flowering pathways occurs on the transcriptional regulation of floral pathway integrators including *LEAFY* (*LFY*), *FLOWERING LOCUS T* (*FT*), and the MADS-box transcription factor gene *SUPPRESSOR OF OVEREXPRESSION OF CONSTANS 1* (*SOC1*) [25,26,27,28,29]. LFY and FT directly activate the MADS-box gene *APETALA1* (*AP1*), while AP1 and LFY promote the expression of other flower meristem identity genes, such as *AGAMOUS* (*AG*) and *SEPALLATA3* (*SEP3*), which specifies the identity of floral meristems and floral organs in Arabidopsis [30]. In Populus trees, two *FT* paralogs control the annual growth cycle. *FT2* is required for vegetative growth and regulates the entry into winter dormancy, while *FT1* is required for bud flush in spring by releasing the dormancy [31]. Similarly, overexpression of the *SOC1*-related Populus gene *MADS12* results in an early bud break in ecodormant *Populus tremula × alba* indicating that *MADS12* expression promotes bud growth reactivation [32]. RNAi-induced simultaneous suppression of all four Populus members of the *AG* subfamily (two *AG* and two *SEEDSTICK* orthologues) in female *Populus alba* results in ‘carpel-inside-carpel’ phenotypes, and female sterility, but did not cause any detectable changes in tree growth or leaf morphology [33]. Therefore, loss of function of the *AG* subfamily in Populus and in Arabidopsis results in similar phenotypes including sterility and floral meristem indeterminacy [34,35]. The overexpression of several transcription factor genes including *FT*, and the MADS-box genes *AG* and *SEP3* causes leaf curling and earlier flowering in Arabidopsis [36,37,38,39]. The function of *AG* and *SEP3* orthologues, such as from lavender (*Lavandula angustifolia*) [40], lily (*Lilium longiflorum, Lilium formosanum*, and Oriental *Lilium* Hybrid) [41,42,43,44,45], orchid (*Oncidium* Gower Ramsey) [43], Japanese gentian (*Gentiana scabra*) [46], *Polypogon fugax* [47], pecan (*Carya illinoinensis*) [48], wheat (*Triticum aestivum*) [49], London plane (*Platanus acerifolia*) [50], peach (*Prunus persica*) [51], and Populus (*Populus tremuloides*) [52], was intensively studied by stably and constitutively expression in Arabidopsis.

Here we present the first functional study of YY1 orthologues from a tree species. We cloned the two YY1 gene copies, *YIN* (*PtYY1a*) and *YANG* (*PtYY1b*), from Populus. Our subcellular localization analysis revealed that the GFP-fusion proteins YIN-GFP and YANG-GFP are mainly localized in the nuclei of Arabidopsis. This finding is in line with the predicted function of YIN and YANG as transcription factors. The expression of *YIN* and *YANG* in Arabidopsis resulted in high expression of *AG* and *SEP3*, which likely caused up-curling of the leaves and earlier flowering. The ectopic expression of *AG* and *SEP3* was accompanied by increased *FT* expression in some transgenic lines, while the expression of the other floral pathway integrators *LFY* and *SOC1* was unchanged. Furthermore, the expression of *YIN* and *YANG* caused similar effects as *AtYY1* overexpression to the ABA response in Arabidopsis [13]. Collectively, subcellular localization analysis and functional characterization in the heterologous species Arabidopsis led to the conclusion that YIN and YANG are functional orthologues of the dual-function transcription factor *At*YY1 in Populus.

## 2. Results

### 2.1. Isolation and Phylogenetic Analysis of YIN and YANG

We searched the published genome data bank of Populus (*Populus trichocarpa*) (https://phytozome.jgi.doe.gov, accessed on 22 September 2021) and found two copies of *YY1*, *PtYY1a* and *PtYY1b*, which we renamed *YIN* and *YANG*. The coding region of *YIN* and *YANG* was amplified without stop-codon by PCR using Populus leaf cDNA as templates. The full-length cDNAs were cloned into the *pGGC000* vector [53] and their sequences were validated by sequencing.

In order to analyze the phylogenetic position of *YIN* and *YANG* within the Rosids clade, and to estimate their gene duplication, we performed a phylogenetic analysis. The phylogenetic tree was constructed with 25 YY1 orthologues from 17 different plant species (Figure 1). Within the Salicaceae family, Populus (*Populus trichocarpa*), *Populus deltoides*, and *Salix purpurea* each have two copies of YY1 clustered in two groups. Group A includes *YIN* (*PtYY1a*), *PdYY1a*, and *SpYY1a*, while group B includes *YANG* (*PtYY1b*), *PdYY1b*, and *SpYY1b*. In contrast, *Ricinus communis* of the Euphorbiaceae, which is a sister family of the Salicaceae within the order of Malpighiales, has only one copy of YY1. This is in line with the assumption that the duplication of *YY1* in the Salicaceae family and clustering in two groups is a result of the “salicoid” WGD that occurred 60–65 million years ago [6]. Although other species of the Rosids clade carry also two or even three YY1 copies, all these duplications occur later and independently of the “salicoid” WGD (Figure 1).

To characterize the phylogeny of the YY1 proteins in more details, we generated a multiple sequence alignment with the six YY1 of the Salicaceae family and *At*YY1 (Figure 2). In all YY1 orthologues, we found the nuclear location sequence (NLS), the casein kinase II (CKII) phosphorylation site, the four Cys2-His2 (C2H2) zinc fingers that likely form a DNA-binding domain, a fifth C2H2 zinc fingers with unknown function, and the acidic C-terminal region that might act as a transcription activation domain [13,20]. The putative C2H2-type DNA-binding domain, the NLS, and the acidic C-terminal region suggest that all YY1 orthologues may function as transcriptional activators. The YY1 orthologues of the Salicaceae family and *At*YY1 share 69.2% overall identity, while in direct comparison *At*YY1 shares 66.2 and 65.6% identity with YIN and YANG, respectively (Table 1). Within the Salicaceae family, the overall identity is 74.2%. These findings indicate that the YY1 orthologues within the Salicaceae family are highly conserved, while sharing with *At*YY1 a high overall identity and all known protein domains.

### 2.2. Expression Patterns of YIN and YANG in Wild-Type Populus Different Tissue

To explore the expression patterns of *YIN* and *YANG* in Populus, we investigated published microarray expression data from the website (phytozome-next.jgi.doe.gov, accessed on 4 August 2022). *YIN* and *YANG* were expressed in all 25 tissues published (Appendix A). Furthermore, we found that in 84% (21 of 25) of the tissues examined, *YIN* was higher expressed than *YANG*. To confirm that *YIN* is the predominately expressed *YY1* orthologue in Populus tissues, we tested *YIN* and *YANG* expression in leaf, stem, and root tissue of Populus by RT-qPCR and found almost identical expression pattern of *YIN* and *YANG* between the published expression data (Figure 3a) and our RT-qPCR results (Figure 3b). In leaf tissue, *YIN* was six-fold higher expressed than *YANG*. Although both *YY1* genes were lower expressed in root, *YIN* was still three-fold higher expressed than *YANG* (Figure 3b). The published expression data reveal that *YANG* was only in late dormant buds two-fold higher expressed than *YIN* (Appendix A). These results indicated that *YIN* is the predominant *YY1* paralogue in most Populus tissues, while *YANG* might have a distinct function particularly in bud dormancy during winter.

### 2.3. Sub-Cellular Localization of YIN and YANG in Arabidopsis and Populus

To localize the YIN and YANG proteins, we generated transgenic Arabidopsis lines (*35S::YIN-GFP* and *35S::YANG-GFP*; hereafter named *YIN_OE* and *YANG_OE*) that expressed YIN and YANG as a fusion protein with GFP, under control of the cauliflower mosaic virus *35S* promoter that allows stable and constitutive expression in plants. Consistent with a role for YIN and YANG in gene regulation, microscopy indicated that YIN-GFP and YANG-GFP were predominantly localized to the nuclei in transgenic Arabidopsis plants (Figure 4). This is in line with their predicted function as transcription factors.

### 2.4. Expression of YIN and YANG Promote Early Flowering in Arabidopsis

To further investigate the functional of *YIN* and *YANG* in plants, we analyzed five independent *YIN_OE* lines (#21, #26, #28, #32 and #34) and five independent *YIN_OE* lines (#6, #7, #11, #12, and #16). All 10 independent T1 transgenic lines showed a 3:1 segregation ratio for kanamycin resistance in the T2 generation, which may indicate a single-copy insertion of the transgenes. Most T3 and T4 plants in all 10 transgenic lines displayed up-curling of juvenile leaves, while most of the later-produced leaves were flat or downward-curled like wild-type leaves (Figure 5). To assess the expression levels of *YIN* and *YANG*, we performed RT-qPCR assays in all 10 transgenic lines and wild-type. As expected, we found high expression levels of *YIN* and *YANG* in 14 days-old-seedlings of all transgenic lines but not in the non-transgenic wild-type plants (Figure 6a). In *YIN_OE* lines, *YIN* were expressed 184.3 to 1745.5 times higher than the wild-type background levels, while in *YANG_OE* lines, expression of *YANG* ranged from 1003.1 to 1895.4 times of wild-type expression (here after named TWE). Although the average expression levels of *YIN* in *YIN_OE* lines variated more, *YANG* expression levels similarly variated within *YANG_OE* lines (e.g., in *YANG_OE #12* between 121.3 and 1885.0 TWE), indicating that the differences in transgene expression levels were not linked to different T-DNA insertions in the independent transgenic lines. Next, we investigated the flowering time for all 10 transgenic lines in comparison with the wild-type control under long-day conditions (Figure 7). A total of 30 days after germination (30 DAG), all plants of the *YIN_OE* and *YANG_OE* lines showed shooting and flower production, while wild-type plants remained in the vegetative phase (Figure 7a–c). During flower development, some petals of the transgenic lines were smaller than wild-type, but neither *YIN_OE* nor *YANG_OE* lines of the T3 and T4 generation displayed homeotic transformation of floral organs as described for *35S::AG* and *35S::SEP3* Arabidopsis flowers (Appendix A) [34,37]. This is in line with our observation that only juvenile and transition leaves of *YIN_OE* and *YANG_OE* showed strong leaf curling, whereas later-produced adult leaves appeared fairly normal (Figure 5d–f). Therefore, the impact of *YIN_OE* and *YANG_OE* on organ morphology decrease along the time axis of the stem [54] and later-produced leaves and flowers are mostly unaffected. In order to analyze the flowering time of the *YIN_OE* and *YANG_OE* lines with more accuracy, we determined their leaf number in comparison to wild-type in four independent experiments with at least 15 plants for each genotype. All *YIN_OE* and *YANG_OE* lines displayed significantly less rosette leaves (ranging between 7.2 ± 0.2 and 7.5 ± 0.1) than the Col-0 control (9.6 ± 0.2) and hence flowered earlier than the wild-type. Notably, we did not find any significant differences of rosette leaf numbers between *YIN_OE* and *YANG_OE* lines indicating that *YIN* and *YANG* expression in Arabidopsis have similar effects on flowering time and hence both putative transcription factors are likely functionally redundant at the protein level.

### 2.5. Expression of YIN and YANG in Arabidopsis Causes High Expression of AGAMOUS and SEPALLATA3

Constitutive expression of several flowering and flower development-related genes, including *FT*, *SEP3*, and *AG*, causes leaf curling and earlier flowering in Arabidopsis [34,35,36,37]. To assess the expression of these endogenous flowering and flower development regulating genes, we performed RT-qPCR assays (Figure 7). First, we tested the expression of the florigen and floral pathway integrator *FT* and found the expression levels of *FT* significantly upregulated in three *YIN_OE* (up to 6.7 TWE) and in three *YANG_OE* lines (up to 6.1 TWE), while four transgenic lines showed no significant differences of *FT* expression in comparison to wild-type (Figure 7b). Next, we tested the expression levels of *SEP3* and *AG* and found both MADS-box genes highly misexpressed (Figure 7c,d). The expression levels of *SEP3* ranged from 22.0 to 68.9 TWE in *YIN_OE* lines and 45.8 to 139.7 TWE in *YANG_OE* lines, while *AG* expression ranged from 158.0 to 453.6 TWE in *YIN_OE* lines and 157.6 to 271.8 TWE in *YANG_OE* lines. We found some a weak correlation between the expression levels of *FT*, *SEP3*, and *AG* in the transgenic lines. The three *YIN_OE* lines with significantly increased *FT* expression (#21, #26, #28), were also the three *YIN_OE* lines with the highest *AG* and *SEP3* expression. Since *FT* was not significantly upregulated in all transgenic lines, we tested also the floral pathway integrators *SOC1* and *LFY* (Figure 7e,f). Surprisingly, neither *SOC1* nor *LFY* expression were significantly changed in any transgenic line tested compared with wild-type. To summarize our expression analysis, only *AG* and *SEP3* were strongly misexpressed in all transgenic lines, while the slight upregulation of *FT* expression in some transgenic lines might be linked to high expression of *AG* and *SEP3*.

### 2.6. Expression of YIN and YANG in Arabidopsis Promotes Root Growth and Seed Germination

Exogenous application of ABA and NaCl can strongly reduce root growth in Arabidopsis plants [55]. Overexpression of *AtYY1* results in enhanced root growth, which is mainly caused by negative regulation of ABA signaling [13]. Our transgenic *YIN_OE* and *YANG_OE* lines also exhibited decreased root growth inhibition sensitivity in response to ABA and NaCl (Figure 8a–d). After one day at 1 µM ABA, the root length of *YIN_OE* and *YANG_OE* plants were unaffected by ABA, while the wild-type root lengths were 83% of its mock control. Importantly, the root length of untreated Col-0, *yy1* mutants, *YIN_OE* and *YANG_OE* plants did not significantly vary at day 1 of treatment. In contrast, NaCl treatment strongly reduced the root length of all genotypes, but *YIN_OE* and *YANG_OE* roots were more resistant than the wild-type. Over the time course (1, 2, and 5 days of treatment), the decreased root growth inhibition sensitivity of *YIN_OE* and *YANG_OE* plants to ABA and NaCl became more evident (Figure 8d). Therefore, expression of *YIN_OE* and *YANG_OE* made Arabidopsis plants less sensitive to ABA and osmotic stress. Similarly, seed germination and cotyledon greening ratio assays can indicate hypo- and hypersensitivity to ABA. *YIN_OE* and *YANG_OE* plants germinated earlier than the wild-type, while the *yy1* mutant control germinated later (Figure 8e,f). To summarize, expression of *YIN* and *YANG* phenocopied the effects of *AtYY1* overexpression on seed germination and root growth, suggesting an overlap of target genes between the transgenic YIN and YANG and the endogenous transcription factor *At*YY1 in Arabidopsis. Since *At*YY1 is a negative regulators of ABA responses, YIN and YANG might have a similar function in Populus.

## 3. Discussion

We isolated from Populus (*Populus trichocarpa*) the full-length cDNAs of two homologous of the Arabidopsis transcription factor gene *AtYY1*, *YIN* and *YANG*, and stably expressed them in Arabidopsis to test their potential protein function as transcription factors. The predicted proteins of *YIN* and *YANG* comprise a nuclear location sequence (NLS), a DNA-binding domain containing four C2H2 zinc fingers, and the acidic C-terminal region that might act as a transcription activation domain (Figure 2). These three features are shared with the Arabidopsis dual-function transcription factor *At*YY1 [13] indicating that *YIN* and *YANG* encode also transcription factors. In line with this assumption, we found YIN and YANG predominantly localized in the nuclei of Arabidopsis (Figure 4). YIN and YANG also contain a fifth C2H2 zinc finger with unknown function and a casein kinase II (CKII) phosphorylation site that is conserved within the plant kingdom [13,20]. Furthermore, YIN and YANG comprise the REPO domain (Figure 2) that is evolutionary preserved and serves as protein–protein interaction domain with the chromatin remodeling complex INO80 that has ATP-dependent nucleosome sliding activity, but also maintains histone H3 levels within the chromatin of target genes [21,22,23,56]. The high amino acid identity between *At*YY1 and YIN and YANG as well as their identical arrangement of protein domains suggest that YIN and YANG have also a high functional similarity to *At*YY1.

Plant orthologues of YY1 function as repressors and activators in diverse processes including mesophyll cell-specific gene repression [18], pathogen responses [19], negative regulation of ABA signaling [13,20], and flowering time control [22]. We found that expression of *YIN* and *YANG* in Arabidopsis accelerates flowering (Figure 7), while the endogenous floral identity genes *AG* and *SEP3* were ectopically high expressed (Figure 6c,d). The MADS-domain transcription factors AG and SEP3 can form a heterotetrameric complex that is essential for the determination of the floral stem cell pool in Arabidopsis, but plays a lesser role in the organogenesis of stamens and carpels [57]. The overexpression of *AG* and *SEP3* causes leaf curling and earlier flowering in Arabidopsis [36,38,39]. Hence, the high expression of *AG* and *SEP3* in *YIN_OE* and *YAN_OE* lines is fully sufficient to explain the leaf curling and early flowering.

Loss of the Polycomb protein CURLY LEAF (CLF) results also in misexpression of *AG* and *SEP3* causing earlier flowering and leaf curling [58,59]. Although *YIN_OE* and *YAN_OE* lines seems to resemble the phenotype of *clf-28* mutants, the dynamics of the leaf curling along the time axis [54] are clearly different (Appendix A). In *clf-28* mutants, the juvenile rosette leaves are only mild curled, whereas the later-produced cauline leaves at the shoot are more strongly curled than the rosette leaves. This is in line with the idea that gradual increasing *SEP3* expression levels (that also occur in wild-type) in later-made leaves enhance the basic misexpression of *SEP3* by loss of *CLF* function. In *YIN_OE* and *YAN_OE* lines, it is rather the other way around with a continuous weakening of the leaf curling along the time axis (Appendix A). This might indicate a gradual vanishing of an essential cofactor for activation of *AG* and *SEP3*. Notably, overexpression of the epigenetic regulator *ULT1* [60] can result in *AG* misexpression causing a *clf*-like phenotype [61] that is considered to indicate Trithorax (TrxG) activity, which antagonizes PcG gene silencing [62]. From there, the phenotype of *YIN_OE* and *YAN_OE* plants could be a kind that YIN and YANG work as TrxG proteins.

During the vegetative rosette stage, the *AG* chromatin is silenced by H3K27 trimethylation, which results in compact and non-accessible chromatin [58,63]. Nevertheless, YIN and YANG can activate *AG* in Arabidopsis, likely by gaining access to the *AG* promoter by either inherent pioneer factor-like activity or by recruiting chromatin remodeler-complexes. In mammalians, binding of YY1 to its DNA binding sites in target genes requires the ATP-dependent chromatin-remodeling complex INO80 that is involved in the shifting and repositioning of nucleosomes, suggesting that YY1 uses the INO80 complex not only to activate transcription but also to gain access to target promoters [23,64]. Recently, it has been shown that *At*YY1 interacts with INO80 and an INO80-associated COMPASS histone H3K4 methyltransferase complex, thereby facilitating the activation mark H3K4 trimethylation and transcriptional activation [22]. Although *YIN_OE* and *YAN_OE* Arabidopsis lines are heterologous systems, it would be interesting to address in future studies whether the reactivation of *AG* by YIN and YANG is based on chromatin remodeling or H3K4 methylation by the INO80 and/or the associated COMPASS complex, respectively. This research could give new insights to how the evolutionary conserved YY1 proteins reactivate target genes in plant and animal systems.

In Populus (*Populus trichocarpa*), the expression pattern of both *AG* orthologues *PtAG1* and *PtAG2* is nearly identical in female and male trees, and consistent with *AG* expression in Arabidopsis, *PtAG1* and *PtAG2* are expressed in stamens and carpels, but not in perianth structures [65]. In contrast to *AG* in Arabidopsis, *PtAG1* and *PtAG2* were also detected in vegetative tissue including stem, leaves, vegetative buds, and vegetative apical meristem tissues [34,65]. Hence, there is a large expression overlap of *YIN* and *YANG* with *PtAG1* and *PtAG2*. However, whether YIN and YANG are direct activators of *PtAG1* and *PtAG2* in Populus need to be investigated in future studies.

Although the gene duplication of YY1 occurred likely about 60–65 million years ago during the early evolution of the Salicaceae, YIN and YANG remain conserved in Populus, suggesting that both gene copies have been under positive selection during evolution and both seem functionally important. In our analyses, we neither found significant differences in the investigated phenotypes nor in expression levels of *YIN/YANG*, *AG*, *SEP3* and *FT* between all 10 transgenic lines (Figure 5, Figure 6, Figure 7 and Figure 8). This is in line with the hypothesis that the protein function is conserved between YIN and YANG by positive selection. However, the limitations of using a heterologous system in this study might miss slight differences in function of YIN and YANG, which could be detectable in Populus.

The positive selection during evolution, and the maintenance of both Populus YY1 genes, could be based on their differential expression pattern (Figure 3 and Appendix A). This different spatiotemporal expression pattern is likely caused by differences in the binding motifs of the promoters of *YIN* and *YANG*. Although YIN is more strongly expressed than YANG in the majority of Populus tissues, they are both expressed and therefore likely contribute to the ‘YY1-function’ accordantly to the ratio of their expression levels. There is one extreme exception from the general dominance of YIN over YANG. In late dormant buds, YANG is more than two times stronger expressed than YIN in any other tissue, indicating an important function of YANG in the late stage of dormant buds, either to maintain the dormancy or more likely to break it. Interestingly, the two Populus *FT* paralogs control the annual growth cycle differently by having opposite roles in the bud development. *FT2* is required for vegetative growth and regulates the entry into winter dormancy, while *FT1* is required for bud flush in spring by releasing the dormancy [31]. It is possible that the high peak of YANG expression in the late stage of dormant buds is related to the function of *FT1* breaking bud dormancy in Populus. However, to prove or disprove such a relationship, and which other functions YIN and YANG might have, requires further studies on the Populus tree, and deeper investigation on a molecular level.

Our work confirmed that YIN and YANG are nuclear-localized proteins that share typical features with the Arabidopsis dual-function transcription factor *At*YY1, which are conserved within the Salicaceae family including a high amino acid identity particularly in the conserved domains that include the REPO domain, a C2H2-type DNA-binding domain, a NLS, and an acidic C-terminal region. These domains are associated with transcriptional regulation, while the acidic region is a typical feature of transcriptional activators. We found that ectopic expression of *YIN* and *YANG* causes high expression of *AG* and *SEP3* leading to early flowering Arabidopsis plants with curled leaves. Since *AG* is silenced during vegetative development of Arabidopsis, the ectopic activation of *AG* in *YIN_OE* and *YANG_OE* Arabidopsis lines indicates that YIN and YANG might function as pioneer factors and/or are recruiters of chromatin-remodeling complexes such as INO80 via their REPO domains. Furthermore, the expression of *YIN* and *YANG* had similar effects as *AtYY1* overexpression to the ABA response in Arabidopsis including earlier seed germination. Together, our results suggest that YIN and YANG are functional transcription factors that regulate gene expression, which likely controls ABA signaling, and possibly flowering time or breaking bud dormancy in Populus. These findings provide new insights into the function of *YIN* and *YANG* in Populus and references toward further investigations on the role of *YY1* genes in woody plants.

## 4. Materials and Methods

### 4.1. Plant Materials and Growth Conditions

In this study, Populus (*Populus trichocarpa*) was cultured on Woody Plant Basal Medium w/Vitamins (WPM [66]; PhytoTech Lab, Lenexa, USA) adjusted to pH 5.8 with 25 g/L sucrose, 5.8 g/L agar and 20 µg/L IBA (Indole-3-Butyric Acid; PhytoTech Lab, Lenexa, USA) under long-day (LD) conditions (16-h-light/8-h-dark cycle; 25 °C) in phytocabinets. The Populus leaf, stem, and root tissues were harvested from four weeks old plants.

The Arabidopsis (*Arabidopsis thaliana*) ecotype Columbia-0 (Col-0) was used as wild-type (WT), *clf-28*, and *yy1* mutants (*yy1-2*, SALK_040806C) were described before [13,19,59,67]. Arabidopsis seedlings were cultivated on 1/2 MS culture medium plates: seeds were stratified for three days at 4 °C in dark and then moved to LD (22 °C). A total of 11 days after germination (DAG), the *Arabidopsis* plants were transferred and continuously grown on soil containing vermiculite and perlite with the ratio of 1:1:1 under LD conditions. For expression analysis (see Section 2.4), 14-day-old seedlings were harvested in two independent experiments.

### 4.2. Multiple Sequence Alignment and Phylogenetic Analysis

We downloaded 25 sequences of *YY1* homologs from 17 different plant species (Appendix A). The plant YY1 protein sequences were obtained from *Phytozome* (https://phytozome.jgi.doe.gov, accessed on 22 September 2021 for *YIN* and *YANG*, and with the exception for Sapur.003G005600.2 (23 March 2023), all other Rosid YY1 genes on 27 May 2022) [68]. To analyze the phylogenetic relationship of YIN and YANG with the other plant YY1 homologs, MEGA7.0.26 software (https://www.megasoftware.net) was employed to create a Neighbor-Joining (NJ) tree [69], and the number of replications in the Bootstrap analysis was set to 1000. The multiple sequence alignment of the salicoid YY1 homologs and *At*YY1 was performed using the DNAMAN software (Lynnon Corp., Quebec City, QC, Canada).

### 4.3. Cloning, Plasmid Constraction and Stable Plant Transformation

The coding region without stop-codon of *YIN* and *YANG* were amplified by PCR using Populus leaf cDNA as templates and gene specific primers. The PCR products were cloned into the *pGGC000* entry vector of the GreenGate cloning system that bases on the Golden Gate technique [53]. After the GreenGate reaction with *pGGZ001* (destination vector), *pGGA004* (*35S* promoter), *pGGB003* (B-dummy), *pGGC000* (CDS of *YIN* or *YANG*), *pGGD001* (*linker-GFP*), *pGGE001* (*RBCS* terminator), and *pGGF008* (*pNOS::BastaR:tNOS*) [53], the recombined *35S::YIN-GFP* and *35S::YANG-GFP* constructs were introduced into the *Agrobacterium tumefaciens* strain GV3101 (*pSoup*) to transform Arabidopsis (Col-0) plants by floral dip method [70]. The transgenic lines were selected by BASTA resistance. The primers used for the cloning of *YIN* and *YANG* are listed in Appendix A.

### 4.4. RT-qPCR Expression Analysis

The detection of gene expression by RT-qPCR were performed as described before [71]. In short, total RNA was extracted using TRIzol Reagent (Invitrogen, Carlsbad, CA, USA), cDNA was synthesized by reverse transcription by the first-strand cDNA synthesis kit (Thermo scientific; Vilnius, Lithuania), and RT-qPCRs were performed using the SYBR Green I Master Mix (Roche Diagnostics, Rotkreuz, Switzerland) and the Roche Lightcycler480 II machine. The expression values were calculated as Mean Normalized Expression (MNE) [72]. For Populus tissue, the *PtActin 7* gene was used as an internal control to normalize the data [73], while for Arabidopsis seedling tissue the *eIF4A1* gene was used [74]. A Student’s *t*-test was used to examine the significance of expression differences between transgenic lines and the Col-0 control. The threshold for statistically significant differences was set to * *p* ≤ 0.05, ** *p* ≤ 0.01, *** *p* ≤ 0.001, and **** *p* ≤ 0.0001.

### 4.5. Subcellular Localization

To analyze the subcellular localization of YIN and YANG, YIN-GFP and YANG-GFP were observed in transgenic *35S::YIN-GFP* or *35S::YANG-GFP* Arabidopsis seeding roots using a confocal laser scanning microscope (LSM 700, Zeiss, Jena, Germany) with a 488 nm excitation laser.

### 4.6. Flowering Time

In four independent experiments (biological repeats), the rosette leaf number of 10 homozygous transgenic Arabidopsis lines (5 YIN_OE lines and 5 YANG_OE lines, all in T4 generation) and Col-0 was determined with 15 or more plants per line in each experiment. Statistical significance was determined by one-way ANOVA followed by a Tukey’s multiple comparisons test. The diagrams and graphs were generated using GraphPad Prism 7 Software (https://www.graphpad.com/scientific-software/prism/, accessed on 29 June 2022) and modified by Adobe Photoshop 8.0.

### 4.7. Seed Germination, Root Length, and Root Growth Assays

For root growth assays, seeds were sowed on MS agar medium, kept at 4 °C for 2.5 days, and then incubated in a growth chamber at 22 °C for 3 days. The resulting seedlings were then transferred to new MS plates (N ≥ 6 plates with approximately 10 plants of each genotype) containing 0 µM (mock), 1 µM ABA, or 150 mM NaCl for 5 days, and the plates were placed vertically on a rack. Root length was measured at 1 day, 2 days, and 5 days of treatment. Relative root growth was calculated as described previously [75].

For seed germination assays, approximately 50 seeds each from Col-0, *yy1* mutant, and trans-genic *35S::YIN-GFP* and *35S::YANG-GFP* plants were planted on MS agar medium with 0.5 µM ABA, 150 mM NaCl, or mock, incubated at 4 °C for 2.5 days and then in a 22 °C growth chamber under long day conditions. Germination (radicle emergence) and cotyledon greening was scored 2 days after incubation.

## Figures and Tables

**Figure 1 ijms-24-07639-f001:**
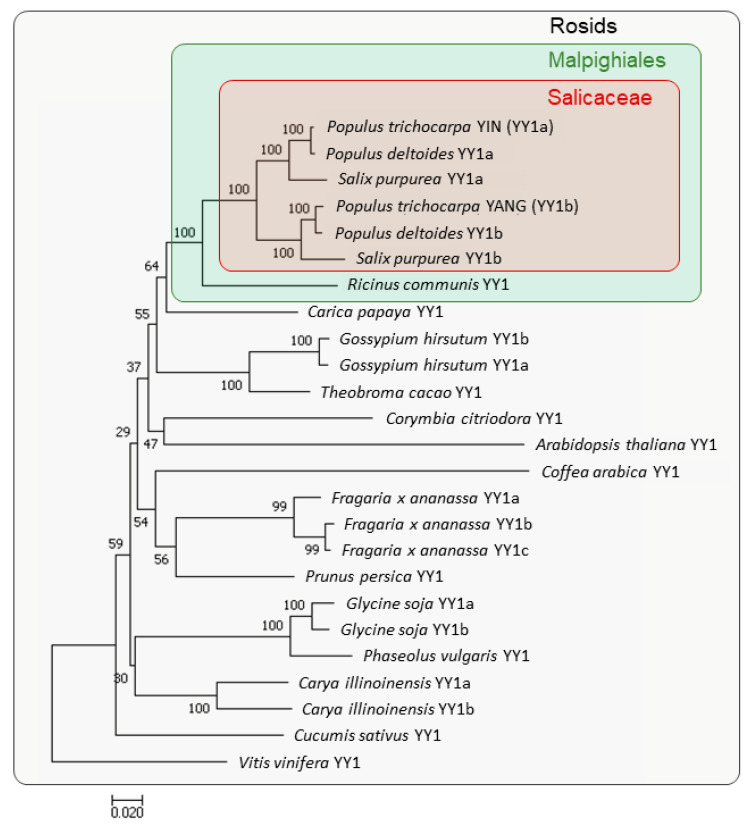
Phylogenetic relationship of YY1 proteins from different plant species of the Rosids clade. The phylogenetic tree constructed with 25 *YY1* genes from 17 different species. The Salicaceae family is marked by a red box, while the Malpighiales clade is marked by a green box. Bootstrap support values are given at the nodes. The phylogenetic analyses were conducted with MEGA 7 using the neighbor-joining (NJ) method and 1000 repetitions of bootstrap tests. All YY1 Sequence were obtained from The Phytozome (https://phytozome-next.jgi.doe.gov, accessed on 27 May 2022 and 23 March 2023, see Section 4.2). Gene IDs with the used versions and the Phytozome Genome IDs are listed in Appendix A.

**Figure 2 ijms-24-07639-f002:**
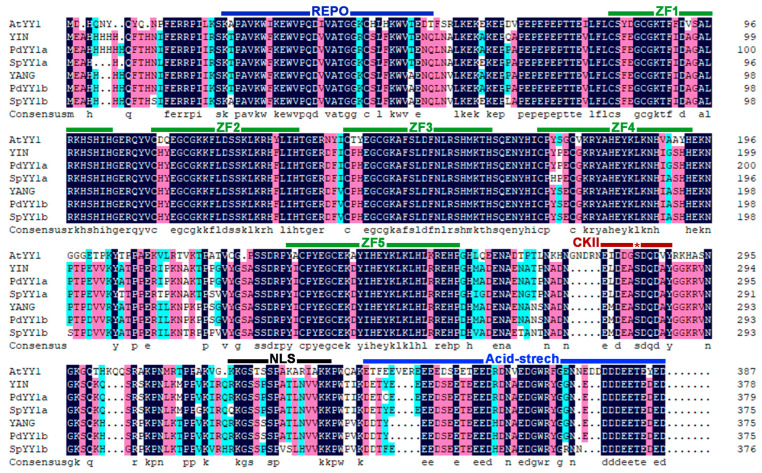
Sequence alignment and prediction of conserved domains in YY1 homologs of the Salicaceae family and *At*YY1. The amino acid sequences were aligned with DNAMAN, modified. The REPO domain [23] is marked in dark-blue, the Zn-finger domains (ZF) are marked by green lines, the CKII phosphorylation site (CKII) by a red line (the asterisk marks the phosphorylated amino acid), the NLS by a black line, and the C-terminal acid-stretch by a line in ultramarine. Identical and similar amino acid residues are shaded with dark-blue, pink, and light-blue, respectively. *At*YY1, *Arabidopsis thaliana*; YIN, YANG, Populus (*Populus trichocarpa*); *Pd*YY1a,b, *Populus deltoides*; *Sp*YY1a,b, *Salix purpurea*.

**Figure 3 ijms-24-07639-f003:**
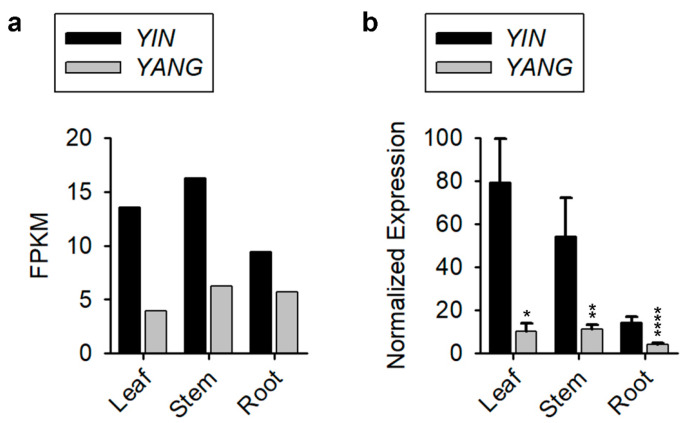
Expression of *YIN* and *YANG* in different tissues of Populus (*Populus trichocarpa*). (**a**) Published expression data of *YIN* and *YANG* in leaf, stem, and root from the Website (phytozome-next.jgi.doe.gov, accessed on 4 August 2022). (**b**) RT-qPCR data of *YIN* and *YANG* expression in leaf, stem, and root of 30 days old Populus (wild-type) plants cultivated in bottles. Data represent the mean ± standard error (SE) from three biological replicates (N = 3), and *PtActin 7* was used as internal control. Note that the plants for the tissue samples continually grew on media with low IBA concentrations. The asterisks indicate significant differences between *YIN* and *YANG* expression (Student’s *t*-test: * *p* ≤ 0.05, ** *p* ≤ 0.01, and **** *p* ≤ 0.0001).

**Figure 4 ijms-24-07639-f004:**
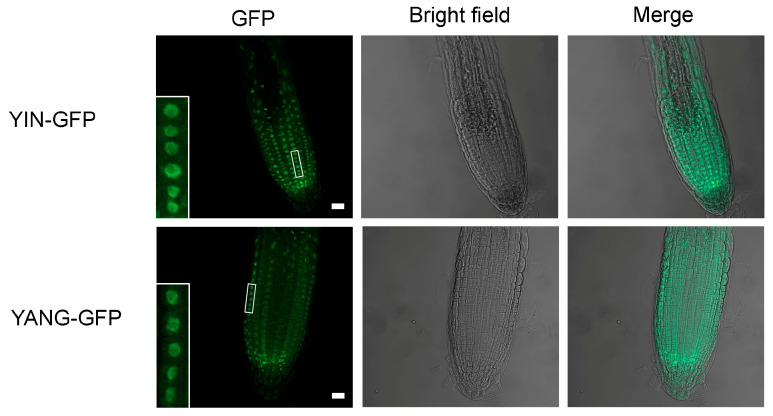
Subcellular localization of YIN-GFP and YANG-GFP fusion proteins in Arabidopsis. Stable transgenic expression of the YIN-GFP and YANG-GFP fusion proteins in Arabidopsis root cells. Note that both fusion proteins are mainly localized in the nuclei (see details in close-ups). Scale bars, 10 μm.

**Figure 5 ijms-24-07639-f005:**
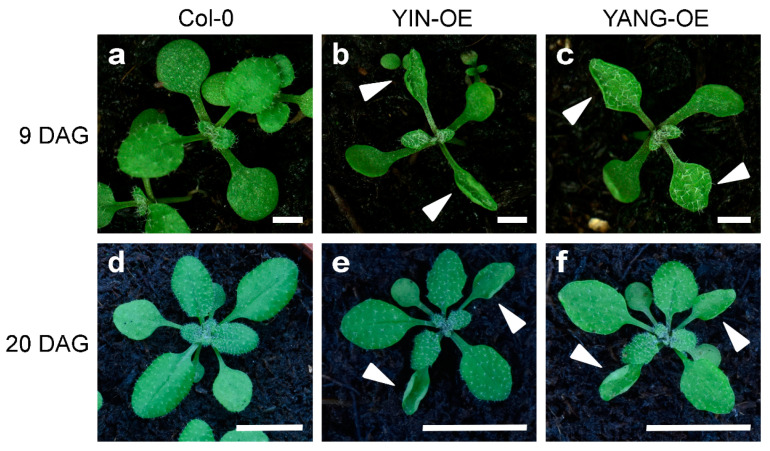
Vegetative phenotype of Arabidopsis *YIN_OE* and *YANG_OE* lines (T4 generation). The plants grew on soil for nine days (**a**–**c**; scale bars = 100 mm) or twenty days (**d**–**f**; scale bars = 1 cm), respectively. The overexpression of *YIN* and *YANG* caused up-curling of the primary leaves (arrowheads). Note that the leaf curling phenotype affected mainly juvenile leaves and was homogeneous mild in the T3 and T4 generation of all 10 transgenic lines (5 *YIN_OE* and 5 *YANG_OE* lines) which were used for all analyses in this study. However, some plants of the T1 generation showed stronger curling in almost all leaves (Appendix A).

**Figure 6 ijms-24-07639-f006:**
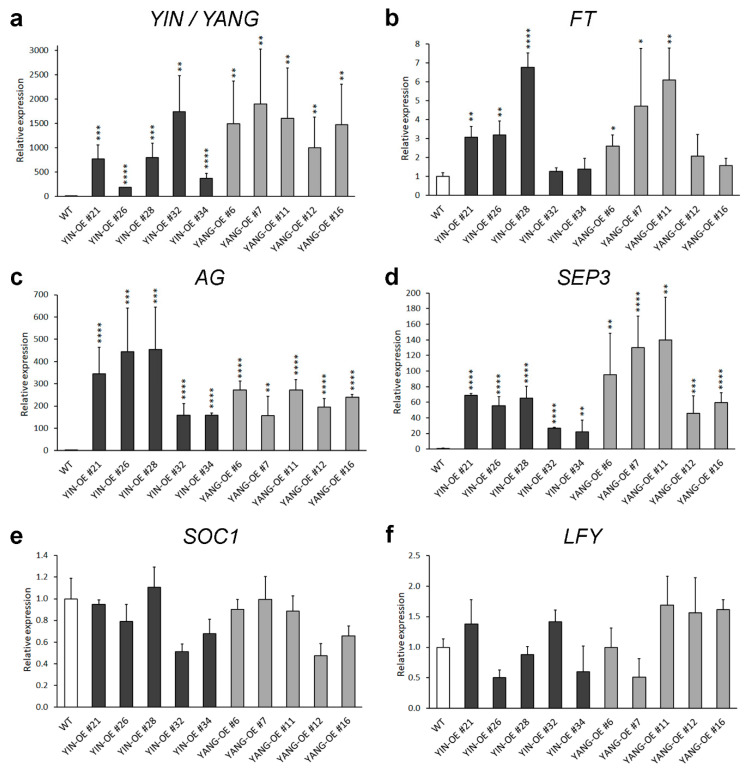
Relative gene expression of *YIN* and *YANG* (**a**) and endogenous flowering-related genes (**b**–**f**) in 14-day-old seedlings of Arabidopsis wild-type (WT, Col-0) and transgenic lines (*YIN_OE* and *YANG_OE*). Dark-gray bars represent independent *YIN_OE* lines (#21, #26, #28, #32, and #34); light-gray bars represent independent *YIN_OE* lines (#6, #7, #11, #12, and #16). The RT-qPCR data represent the mean ± standard error (SE) from at least two biological replicates per transgenic line (N ≥ 2), and ten biological replicates for Col-0 (N = 10). *eIF4A1* was used as internal control, and the expression levels in the Col-0 control were set to 1. The asterisks indicate significant differences compared with the Col-0 plants (Student’s *t*-test: * *p* ≤ 0.05, ** *p* ≤ 0.01, *** *p* ≤ 0.001, and **** *p* ≤ 0.0001). Note the different expression scale ranges of the tested genes. Furthermore, note that there were no significant changes between the expression of *SOC1* and *LFY* in the transgenic lines and their expression in wild-type (Col-0).

**Figure 7 ijms-24-07639-f007:**
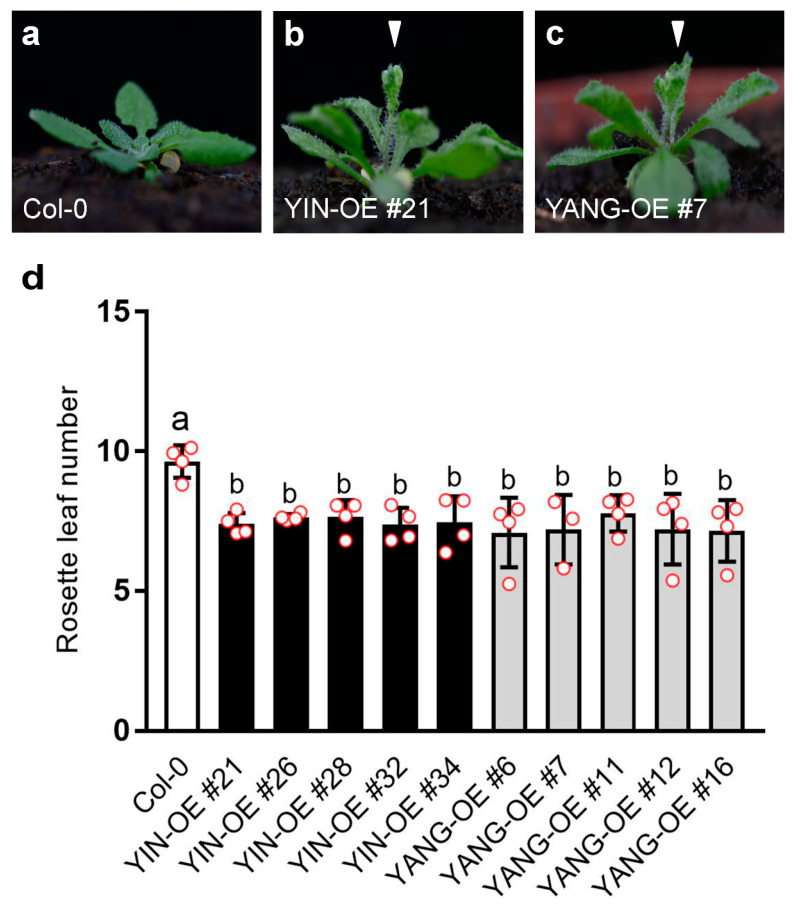
Expression of *YIN* and *YANG* accelerates flowering in Arabidopsis. (**a**–**c**) Arabidopsis plants, 30 DAG under the long-day conditions. *YIN_OE* (**b**) and *YANG_OE* lines (**c**) bolt earlier (arrowheads) than the wild-type control (**a**, Col-0). (**d**) Rosette leaf number in the wild-type and transgenic *YIN_OE* and *YANG_OE* lines. The rosette leaf number was measured in four independent experiments (biological replicates; N = 4; *YANG_OE #7*, N = 3). Red dots indicate the average in each biological replicate with at least 15 plants (N ≥ 15). Error bars represent the standard deviations. Statistical significance was determined by one-way ANOVA followed by a Tukey’s multiple comparisons test.

**Figure 8 ijms-24-07639-f008:**
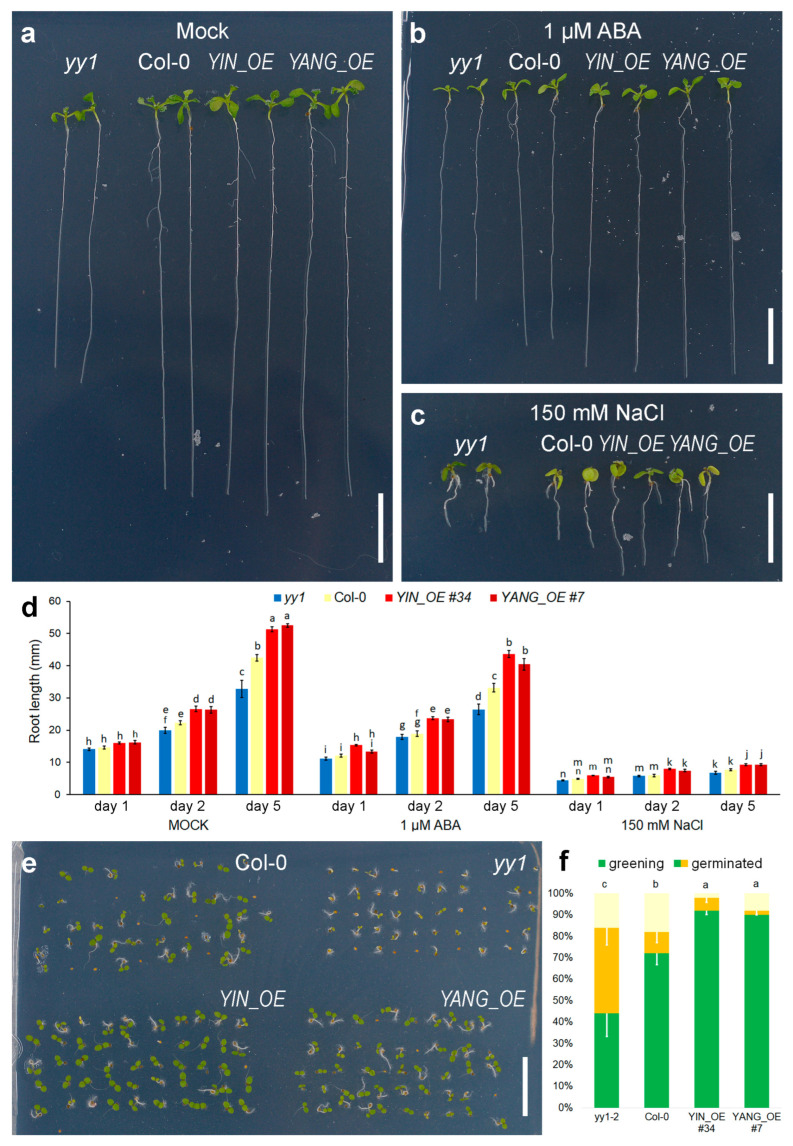
Expression of *YIN* and *YANG* promotes root growth and accelerates seed germination in Arabidopsis. (**a**–**d**) Root growth among *yy1*, Col-0, and transgenic *YIN_OE* (*#34*) and *YANG* (*#7*) plants. Three-day-old seedlings grown on MS medium were transferred to MS medium containing 0 (**a**, Mock), 1 µM ABA (**b**), or 150 mM NaCl (**c**), and grown vertically for 5 days. Scale bars = 10 mm. (**d**) Root length in mm; N ≥ 6 plates with approximately 5–10 plants of each genotype. (**e**,**f**) Seed germination and cotyledon greening ratios of Col-0, *yy1*, and *YIN_OE* (*#34*) and *YANG* (*#7*) transgenic plants (N = 50). Seeds geminated on MS medium for 4 days. (**d**,**f**) Error bars represent the standard error of the mean; statistical significance (*p* ≤ 0.05) was determined by Student *t*-test and a-n mark groups of significant differences.

**Table 1 ijms-24-07639-t001:** Similarity and identity of *At*YY1, YIN, and YANG.

	***At*YY1**	**YIN**	**YANG**	** Identity **
***At*YY1**		66.2%	65.6%	
**YIN**	81.1%		92.0%	
**YANG**	79.9%	94.1%		
	** Similarity **			

## Data Availability

All relevant data are included within this article. More detailed information on the phylogenetic analysis is available on request to ralf.mueller@hhu.de.

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
