# Peer review of "Expression of the Populus Orthologues of AtYY1, YIN and YANG Activates the Floral Identity Genes AGAMOUS and SEPALLATA3 Accelerating Floral Transition in Arabidopsis thaliana"

_ijms, 2023, doi:10.3390/ijms24087639_

Round 1

Reviewer 1 Report

The manuscript is overall clear but many details are missing in the document, which prevents replication and fully understanding what was done. I find it odd that all data is presented within this study (as stated by the authors) when that is not the case. The authors have produced new genetic data that should be deposited in a public repository. The matrix used to build the tree is also not accessible – thus, there is a lot of raw data that as a reviewer I did not have access to. Other comments are:

Scientific comments:

- Please clarify the number of samples/plants or replicates used in each experiment. It is very hard to follow how many were used.

- Please explain why homologs were searched in those 17 species (and not others).  And why did the NJ method was chosen? This is too limited to understanding gene duplication as stated in the manuscript. The BS values indicate in the tree are also too low to make conclusions about this statement. Also note that in this section, results are mixed with discussion.

- The results of Table 1 are also indicated in the text. So, this is redundant to me.

- Comparison of results with expression data already published should be better explained. Conditions often vary between studies and plants, and thus results might be biased because of this.

- Please indicate the exact p-value in the results. Also, note that an ANOVA is indicated in (some) results while a student text is indicated in M&Ms. Letters indicating differences are missing in Figure 6. Concerning this figure,  I don’t follow why there are no differences in the genes +resent in e, and f. In relation to Figure 7, I don’t follow the red dots, but those were clearly manually edited/added. Thus, I don’t recommend its use.

Proofreading comments:

-          Please ictalize species names throughout the document (including the tree).

-          Please ictalize gene IDs throughout the document.

-          Figure 2b should be 3b.

-          There are some minor grammatical mistakes in the text that would benefit from text editing.

Author Response

Dear Reviewers,

>> We would like to thank you for the positive feedbacks and the constructive suggestions that helped us to further improve our manuscript.

>> In the following we are going to comment on every point raised. Importantly, to address major concerns of Reviewer 1 and Reviewer 2, we re-run our phylogenetic analysis of YIN and YANG and created a new phylogenetic tree. As reviewer 2 suggested, we used Vitis vinefera YY1 as outgroup. We hope that you find our new phylogenetic analysis more convincing.

>> We modified Figure 1 and Figure 2, added more information to the main text and figure legends as suggested by the reviewers, extended the discussion as suggested by reviewer 3, and added the new Supplementary Figure 3 with leaf curling phenotypes. All these changes are reflected in some minor changes in the main text of the results and discussion.

>> Please note that smaller changes such as grammar and spelling corrections are here not listed, but marked using the “Track Changes” function in the revised manuscript.

REVIEWER 1:

The manuscript is overall clear but many details are missing in the document, which prevents replication and fully understanding what was done. I find it odd that all data is presented within this study (as stated by the authors) when that is not the case. The authors have produced new genetic data that should be deposited in a public repository. The matrix used to build the tree is also not accessible – thus, there is a lot of raw data that as a reviewer I did not have access to.

>> We are extremely grateful for pointing out the problem with our phylogenetic analysis and tree. We therefore re-run our phylogenetic analysis and created a new phylogenetic tree (see below). We are very thankful for the suggestions. For more clearly and in accordance with your concerns, we have added more detailed information in Supplementary Table S1: We added the Plant Family, the Gene ID with the used version (e.g. AT4G06634.1), and the Phytozome Genome ID/version (as requested by reviewer 2) with internet link so that the readers can easily retrace which protein sequences were used in our phylogenetic analysis. If you or one of the other reviewers would like to see our detailed analysis and results of the new phylogenetic analysis in more detail, we would be glad to provide all data to you. However, in our opinion, it is unusual to deposit this type of data in a public repository. However, we changed the Data Availability Statement and add "More detailed information on the phylogenetic analysis are available on request to [email protected]." The Legend of Figure 1 reads know "The phylogenetic analyses were conducted with MEGA 7 using the neighbor-joining (NJ) method and 1000 repetitions of bootstrap tests. All YY1 Sequence were obtained from The Phytozome (https://phytozome-next.jgi.doe.gov). Gene IDs with the used versions and the Phytozome Genome IDs are listed in Supplementary Table S1."

Other comments are:

Scientific comments:

- Please clarify the number of samples/plants or replicates used in each experiment. It is very hard to follow how many were used.

>> Thank you for pointing out that it might hard to follow how many samples were used for some readers. Although we gave the number of biological replicates in writing (e.g. in lines 196-197 "Data represent the mean ± SE from three biological replicates, ..."), we add now the 'N-numbers'. Lines 196-197 read now " Data represent the mean ± standard error (SE) from three biological replicates (N = 3), ..."). We added also the 'N-numbers' in the lines: 256, 257, 296, 297, 328 and 329. Additionally, we double checked other sample size relevant information such as number of plants, and added information if appropriated. For example the lines 296-297 read now " The rosette leaf number was measured in four independent experiments (biological replicates, N = 4; YANG_OE #7, N = 3). Red dots indicate the average in each biological replicate with at least 15 plants (N ≥ 15)."

- Please explain why homologs were searched in those 17 species (and not others). 

>> Thank you for your comment. As we wrote in line 123-126 "In order to analyze the phylogenetic position of YIN and YANG within the Rosids clade, […] The phylogenetic tree was constructed with 25 YY1 orthologues from 17 different plant species (Figure 1)." We searched on the Populus phytozome website for YIN and YANG Homologs, and chose the one with a similarity greater than 80%. But in the end, our choices of roside species were led by the availability of YY1 sequences (phytozome website) and the common choices in other publications. We do not think that any backwards rationalization of our choices would be suitable for our article.

- And why did the NJ method was chosen?

>> We chose the NJ method, since it is one of the standard methods to build phylogenetic trees. However, we reconstructed also a second phylogenetic tree by the Maximum likelihood (ML) method but the result was less convincing (the bootstrap support values were lower) and we therefore sticked to the NJ method.

- This is too limited to understanding gene duplication as stated in the manuscript. The BS values indicate in the tree are also too low to make conclusions about this statement.

>> Sorry for the confusion caused by our original phylogenetic analysis. In our new generated phylogenetic trees the BS values are much higher. We hope that allays your concerns.

- Also note that in this section, results are mixed with discussion.

>> We fully understand your point of view. However, in our opinion, any part of a scientific article have to be understandable without reading the other parts. Therefore, also the result part need some conclusions, and sometimes a short discussion, so that the results are understandable for the readers (without reading the other parts of the article).

- The results of Table 1 are also indicated in the text. So, this is redundant to me.

>> We fully understand your point of view. Nevertheless, we think that the redundancy will help the readers to understand the results and the article. Once, a reviewer had exactly the opposite point-of-view: in an article, we wrote the values of an experiment only in the main text, but the reviewer complained that we did not refer to any table or figure.

- Comparison of results with expression data already published should be better explained. Conditions often vary between studies and plants, and thus results might be biased because of this.

>> We are sorry to learn that you think that here could be any bias. However, in our opinion, this is not correct since: (i) If conditions vary between studies and plants, these should rather produce different and not similar results as we found. (ii) We could not know whether our expression analysis (RT-qPCR) would confirm or "rebuttal" the published expression data. Therefore, we cannot see any bias in our experimental design nor in our results. However, if you still think that there was bias, please explain it to us so that we can discuss and point to the problem in our article so that the readers are aware that there could be a problem with bias.

- Please indicate the exact p-value in the results.

>> In our opinion, exact p-value are absolutely rare in graphic figures of scientific publications (since they would make the graphics hard to read).

- Also, note that an ANOVA is indicated in (some) results while a student text is indicated in M&Ms.

>> We used ANOVA only once, for the flowering time analysis. This ANOVA is mention in exactly two sentence: Line 298-299 (Legend of Figure 7) "Statistical significance was determined by one-way ANOVA followed by a Tukey’s multiple comparisons test." and Line 459-460 (Materials and Methods, 4.7 Flowering time) "Statistical significance was determined by one-way ANOVA followed by a Tukey’s multiple comparisons test." Statistical significance was determined by Student t-test in all other experiments of the manuscript, and this is indicated in the figure legends and Materials and Methods.

- Letters indicating differences are missing in Figure 6. Concerning this figure, I don’t follow why there are no differences in the genes +resent in e, and f.

>> We guess that you meant asterisks since Figure 6 do not have letters for indicating significance. However, there was no significant changes in SOC1 and LFY expression (Figure 6e and 6f).

Here we are listing the exacted p-values.

Figure 6e (SOC1): YIN_OE lines #21 (p = 0.92), #26 (p = 0.68), #28 (p = 0.84), #32 (p = 0.35), #34, (p = 0.54) YIN_OE lines #6 (p = 0.85), #7 (p = 0.99), #11 (p = 0.83), #12 (p = 0.12), #16 (p = 0.70)

Figure 6f (LFY): YIN_OE lines #21 (p = 0.37), #26 (p = 0.20), #28 (p = 0.75), #32 (p = 0.28), #34, (p = 0.35) YIN_OE lines #6 (p = 0.99), #7 (p = 0.99), #11 (p = 0.15), #12 (p = 0.23), #16 (p = 0.23)

In relation to Figure 7, I don’t follow the red dots, but those were clearly manually edited/added. Thus, I don’t recommend its use.

>> As we wrote in Materials and Methods: "The diagrams and graphs were generated using GraphPad Prism 7 Software (https://www.graphpad.com/scientific-software/prism/)" but we removed now "and modified by Adobe Photoshop 8.0." since it was simply indicating that we add letters that show significance.

Proofreading comments:

-          Please ictalize species names throughout the document (including the tree).

>> We 'italicize' all Latin species names (binomial nomenclature) in the text and the phylogenetic trees.

-          Please ictalize gene IDs throughout the document.

>> In our opinion, italicized gene IDs are not standard in IJMS, and we do not recall any journal, there this is the case.

-          Figure 2b should be 3b.

>> Changed.

-          There are some minor grammatical mistakes in the text that would benefit from text editing.

>> We carefully read the text and corrected grammatical and other minor mistakes. For details, please see the marked changes using the “Track Changes” function in the revised manuscript.

REVIEWER 2:

Comments and Suggestions for Authors

Dear Author,

         Overall the manuscript is well-planned and properly executed. However, I have some concerns on the few things:

Line 14: YIN was stronger expressed -> modify to YIN was strongly expressed

>> We agree, although "stronger" is grammatically correct, "MORE strongly" sounds better, it reads now "YIN was more strongly expressed than YANG.

Line 149: "lakota" must be a subsp. or variety name. It can be removed.

>> Removed.

In figure, it is commonly oberseved less than 30 bootstrap on distantly related species. How author can justify 5, 9, 18, 26 in the branches. When we concluding the results based on the evolution of plant species. Are those values significant? Arabidopsis represent as "outgroup" than Vitis vinefera. For Eurosids, Vitis must be an outgroup. It is not observed in case of YY1a and YY1b. Why so?

>> We are extremely grateful for pointing out the problem with our phylogenetic analysis and tree. We therefore re-run our phylogenetic analysis and created a new phylogenetic trees (see above, our response to reviewer 1, who had similar concerns). We think that you was exactly right and the problems occurred from choosing the wrong outgroup. Thank you very much again for pointing to the outgroup problem and to suggest Vitis as outgroup.

In Figure 2: ZF3 was observed with insertion in SpYY1b. Is it could be an annotation error? Have author checked with another genome version?

  >> Thank you for pointing out the problem. In the original version we used for Salix purpurea YY1b Sapur.003G005600.1. Now, we made a new alignment (Figure 2) with Sapur.003G005600.2 that do not have the 'insertion'. Additionally, we marked in the new version of the alignment the REPO-like domain, since it strengthens our extended discussion suggested by Reviewer 3.

No significance marks in SOC1 and LFY genes expression.

 >> Since both reviewers, 1 and 2, were wondering about the absence of significance marks in SOC1 and LFY genes expression, we add in the figure legend, line 266: "Furthermore, note that there was no significant changes between the expression of SOC1 and LFY in the transgenic lines and their expression in wild-type (Col-0)." Please note that the p-values (Student t-test) of the Figure 6e (SOC1) and 6f (LFY) are listed above, in our response to Reviewer 1, who had the same concerns.

Many lines in results have citations. like 271, 272, 303,. Can it be avoidable?

>> We fully understand your point of view. However, in our opinion, any quotation of published information demands to cite the references.

Line 322: Populus can be P.

>> Thank you for pointing out the inconsistency. However, we prefer the 'English name' Populus as species name and removed all 'P. trichocarpa' from the manuscript. There 'Populus' could confuse the readers, e.g. comparison with other Populus species like Populus deltoides, we use 'Populus (Populus trichocarpa)'. 

Line 336: Cys2-His2 cab be C2H2

>> DONE.

 Line 403: Here author used IBA, YIN/YANG expression was influenced by auxin's level. This experiment could have conducted either in basal media or ex vitro conditions. This point is more critical, for tissue specific expression gene must be analysed in normal condition.#

>> We agree that the low IBA levels could affect YIN/YANG expression levels, but the expression patterns of our RT-qPCRs (Figure 3b) and the published expression data (Figure 3a) are similar, which make it very unlikely that the differences between the expression of YIN and YANG are caused by IBA (the different expression levels of YIN and YANG are to some extant the whole point of the expression analysis). However, we added in the figure legend the sentence 'Note that the plants for the tissue samples continually grew on media with low IBA concentrations.'

Line 406-409: seedlings was cultivated or seeds were stratified under 4C in 1/2 MS media. 

>> Changed.

Line 416: Versions of genomes. All from same database?

>> Yes, all sequence data are from phytozome-next.jgi.doe.gov. We added the used genome versions in Supplementary Table 1.

Finally, conclusion could have given with clear new insight about possible mechanism(s) of YIN/YANG in Populus.  

>> We split the last sentence of the discussion in two and add possible functions of YIN/YANG in Populus, the two sentence reads now  "Together, our results suggest that YIN and YANG are functional transcription factors that regulate gene expression, which controls likely ABA signaling, and possibly flowering time or breaking bud dormancy in Populus. These findings provide new insights into the function of YIN and YANG in Populus and references toward further investigations on the role of YY1 genes in woody plants." Additionally, we included some points about possible mechanism(s) in the discussion of the revised manuscript (please see there).

REVIEWER 3:

Comments and Suggestions for Authors

Overall, I found the article well-written and sound. The conserved role of the dual role transcription factor YY1 in plant development is well explained.

I have a couple of points to put through for the authors.

  1. Would it be possible to show the negative regulation of genes to show the duality of its transcription role? I see the positive effect on flowering time genes.

>> Thank you to point this out. Indeed our study would be more complete if we could show direct negative regulation of YY1 target genes. Unfortunately, the only published direct YY1 targets are only expressed after fungi infection (Lai et al., 2014). We neither have the fungi nor are we willing to risk the use in our lab.

  1. In the discussion, it would be nice to discuss the plausible different roles of YIN and YANG considering the expression pattern in different tissues and the difference in their respective promotors. From the perspective of evolution, why would the duplicated gene still be there?

>> Thank you for your suggestions and we added two long paragraph in the discussion considering from the perspective of evolution (i) the function of YIN and YANG and whether our data support the idea of maintaining equal protein function in both orthologues or not, and then (ii) discuss the plausible different roles of YIN and YANG considering the expression pattern in different tissues. We made further changes to the discussion, also inspired by the comments of the other reviewers. Please see the extended changes in the discussion of the revised manuscript.

We would like to thank all reviewers again for their constructive comments.

Kind Regards,

Ralf Mueller-Xing

Reviewer 2 Report

Dear Author,

         Overall the manuscript is well-planned and properly executed. However, I have some concerns on the few things:

Line 14: YIN was stronger expressed -> modify to YIN was strongly expressed

Line 149.: "lakota" must be a subsp. or variety name. It can be removed.

In figure, it is commonly oberseved less than 30 bootstrap on distantly related species. How author can justify 5, 9, 18, 26 in the branches. When we concluding the results based on the evolution of plant species. Are those values significant ? Arabidopsis represent as "outgroup" than Vitis vinefera. For Eurosids, Vitis must be an outgroup. It is not observed in case of YY1a and YY1b. Why so ?

In Figure 2: ZF3 was observed with insertion in SpYY1b. Is it could be an annotation error ? Have author checked with another genome version ?

No significance marks in SOC1 and LFY genes expression.

Many lines in results have citations. like 271, 272, 303,. Can it be avoidable ?

Line 322: Populus can be P.

Line 336: Cys2-His2 cab be C2H2

 Line 403: Here author used IBA, YIN/YANG expression was influenced by auxin's level. This experiment could have conducted either in basal media or ex vitro conditions. This point is more critical, for tissue specific expression gene must be analysed in normal condition.

Line 406-409: seedlings was cultivated or seeds were stratified under 4C in 1/2 MS media. 

Line 416: Versions of genomes. All from same database ?

Finally, conclusion could have given with clear new insight about possible mechanism(s) of YIN/YANG in Populus.  

Author Response

(The authors gave the same response as above.)

Reviewer 3 Report

Overall, I found the article well-written and sound. The conserved role of the dual role transcription factor YY1 in plant development is well explained.

I have a couple of points to put through for the authors.

1. Would it be possible to show the negative regulation of genes to show the duality of its transcription role? I see the positive effect on flowering time genes.

2. In the discussion, it would be nice to discuss the plausible different roles of YIN and YANG considering the expression pattern in different tissues and the difference in their respective promotors. From the perspective of evolution, why would the duplicated gene still be there?

Author Response

(The authors gave the same response as above.)

Round 2

Reviewer 1 Report

The authors have addressed all previous comments.